# Highlighting the Drivers of Italian Diversified Farms Efficiency: A Two-Stage DEA-Panel Tobit Analysis

Luca Romagnoli, Vincenzo Giaccio *, Luigi Mastronardi and Maria Bonaventura Forleo

Department of Economics, University of Molise, 86100 Campobasso, Italy; luca.romagnoli@unimol.it (L.R.); luigi.mastronardi@unimol.it (L.M.); forleo@unimol.it (M.B.F.)
* Correspondence: giaccio@unimol.it; Tel.: +39-0874-404-404

**Abstract:** Farm diversification is an important phenomenon in agricultural systems and rural development in Europe, pursuing economic, social and environmental goals. For the sustainability of diversified farms, it is important to analyse some drivers affecting farm efficiency, for instance, socio-economic, technical and policy drivers. The efficiency performance of a panel of Italian farms practising other gainful activities in the period 2012–2017 was investigated and regressed against the drivers that mostly affects farm performances. FADN data and a two-step approach were used. An output-oriented Data Envelopment Analysis was applied; in the second step, efficiency scores were used as a dependent variable in a panel Tobit regression analysis used to determine differences in the significance of drivers. Social, economic, technical and policy drivers were considered as explanatory variables. Results show margins for improving farms performances. The incidence of the output from other gainful activities has been proven to positively affect farms efficiencies, while intermediate costs are the most negatively impacting factor. As regards policy variables and implications, the significance of localization in mountain disadvantaged territories further supports the relevance of EU subsidies in less-favoured areas. Managerial implications in terms of technical, structural and economic indicators can be drawn from study findings.

**Keywords:** other gainful activities; less favoured areas; public policies; FADN; regression model

## 1. Introduction

Farm diversification is an important phenomenon in agricultural systems and rural development all over Europe and it is considered a method of integrating farm incomes alongside other profitable activities. Other on-farm income opportunities and the performance of units in terms of efficiency derived from different drivers, such as technology, labour and land use, as well as entrepreneurial characteristics [1]. At the same time, the economic performances of farms are strongly influenced by public financial intervention. In fact, the contribution of diversified farms to the economic and social life in farms and rural areas tend to justify the legitimacy of subsidies [2]; furthermore, its beneficial effects on landscape and other natural resources are well known [3]. This phenomenon has greatly affected Italian farms, whereby 76% of them are presently engaged in other gainful activities [4].

In previous decades, European Union agricultural and rural development policies have been strongly directed towards enhancing the competitiveness and efficiency of farms. The EU financial support for diversification of farms and rural development is quite evident in the generational turnover and in facilities for female entrepreneurship, as well as in the mechanization and better distribution of family work on farms.

In assessing the efficiency of farms with Other Gainful Activities (OGA), particular attention should be devoted to some potential drivers from the farm's context—such as some technical, economic and social drivers—and from the external context, among which is the policy support by the EU Common Agricultural Policy (CAP). CAP supports farm

income via a broad set of different measures, of which the two major strands are for direct income support (First Pillar) -mostly via direct payments, and for rural development (Second Pillar)- mainly through farm investment support, agri-environmental scheme payments, economic diversification in rural areas, and Less Favoured Areas (LFAs) payments. The location of farms in LFAs determines their income and results in higher production costs; despite other activities performed by farmers to diversify and increase farm income, it could remain insufficient for profitable agricultural production in disadvantaged areas, as such justifying subsidies in these areas, especially in the mountain zones.

The study has a twofold aim: 1. assess the efficiency of Italian diversified farms; 2. detect the factors that mostly affects farms efficiency performances, considering the socio-economic, technical and policy variables.

The paper is structured as follows: Section 2 presents a review of current literature on the topic; Section 3 describes the dataset and methods employed in the analysis; Section 4 reports all the results, together with a discussion on the findings; finally, Section 5 concludes the paper with some final comment and directions for future research.

## 2. Literature Background

A rich body of literature has addressed determinants of farm diversification and the impact of socio-economic variables on the efficiency performances of farms, although, results are not always consistent (e.g., among the most recent [5–8]).

As for the farmer's age, it is a cliché to believe that this variable has a negative impact on the financial and economic performances since older farmers are usually less risk-averse and oriented towards investing in diversified and more profitable activities than the younger farmers [9]. However, this is one of the reasons for the strong dedication of policymakers to generation turnover in farms. The gender's impact on diversified farm efficiency is also a debated topic in literature [10].

The role of irrigation in farm diversification is considered relevant because of the important role it plays in determining the land capital and the profitability of farms [11]. A large body of literature has investigated the effects of irrigation development and its impact on income diversification, above all in rural and less-favoured areas, if irrigation systems are feasible [12].

Overall, mechanization is a strategic factor influencing efficiency changes, where farms with less efficiency are more likely to experience greater improvements. The literature points out that energy efficiency in areas with high agricultural intensity is relatively low and therefore, the margins for improvement in terms of technical and economic efficiency are greater in diversified agriculture than in traditional agriculture [13]. The impact of mechanization is significantly positive on the farmers' income [14]. The efficiency of agricultural machinery in agricultural production is much higher than that of human labour, and the cost of production is lower than the cost of manual labour. In other words, the greater the total power of agricultural machinery and the degree of mechanization, the lower the manpower required. It follows that the reason for supporting the mechanization of modern agriculture is to improve the efficiency levels of agricultural holdings [15]. Under a different perspective, machinery-sharing arrangements are found to have a positive and statistically significant impact on farm efficiency of Swedish crop and livestock farms [16].

Literature highlights that a better allocation of the workforce on the farm represents a strategic choice for the entrepreneur, especially if the strong incidence of family work is considered. The choice to diversify farm outputs is often an attempt to solve this problem [9]. The reallocation of intensive labour from traditional agricultural activities on the farm to other farm activities is also a strategy to improve farm profitability or to maintain a parity income level [17]. The ratio of total work units represented by family work is a constant in studies concerning the farm's efficiency, particularly for farms with diversified activities. In this context, literature highlight that surplus labour characterizing specialised farms could be re-deployed to other economic activities well represented in diversified farms. In other words, in addition to its risk-reducing benefits, diversification

offers the opportunity to exploit the complementary potential of the farm and a more efficient distribution of the family workforce, at the same time, minimizing the economic and entrepreneurial risk [18]. Moreover, the percentage of family work is often negatively associated with efficiency [19]; different results were obtained by Sabasi et al. [20] that found a negative association between the (increase in) off-farm work by the farm household and the (decrease in) the efficiency of U.S. dairies.

Among definitions of farm diversification, in addition to the number of activities generating farm output, literature considers the increase in the added value, where it is the income used to measure the land and labour productivities. The literature agrees that specialization in agriculture enables the reduction in unit costs through an increase in the scale of production and, consequently, the increase in the Value Added from agricultural activities; at the same time, this specialization involves a reduction of production costs, such as intermediate consumptions and work units [21]. Thanks to a diversity of incomes, the relevance of diversified activities on Value Added by farms, can be summarised in the dispersion of risk derived from fluctuations in prices on different agricultural markets, and above all, in times of economic downturn [22]. On the other hand, farm diversification could produce a reduction of factor productivities related to an increase in total costs as each activity has its own specific transaction costs [23].

Lakner et al. [24] showed that the expansion of non-agricultural activities on a farm reduces the costs of intermediate inputs in agriculture, as well as the capital defined as the depreciation value of machines and buildings, because of a reduction in core agriculture. At the same time, diversification enables the less intensive use of land.

As regards farming systems, farms' orientation toward specialised or mixed production could be another relevant driver of efficiency performance. From the literature review of recent years about the evaluation of the economic efficiency of animal production, we can deduce that the efficiency of livestock drafts a picture that is not so straightforward. According to some studies, farms specializing in cattle breeding are fully efficient [5]. However, the contribution of animal production to farm performances is related to the kind of livestock present on the farm [25]. Furthermore, studies have used different specific economic indicators, such as total investment and costs, value of production, total profit and profit for livestock units [26]. The transition from crop or livestock production to mixed production is often related to the need to reduce business risk. However, farms with multiple productions find it more difficult to increase their productivity, especially land productivity [27]. In turn, farms running mixed production reveal a lower technical efficiency than specialized crop farms [28]. From the point of view of economic efficiency, some literature affirms that mixed farms have a lower level than specialized farms, due to inefficient use of inputs [29,30]. Farmers who opt for the diversification of income on the farm are the most likely to choose mixed production systems (crop and livestock) and this becomes almost a necessity for farms located in disadvantaged areas or which are family-run, considering the objective of optimizing the use of family work [31].

Some scholars have investigated the impact of subsidies on farms economic performance [32–35]. Indeed, theoretical and empirical studies addressed the topic, though theoretical results on this subsidy–efficiency link are ambiguous and empirical studies also seem inconclusive. A meta-analysis of empirical results about the impact of public subsidies on farm efficiency [36] shows that subsidies are usually negatively associated with efficiency, but the direction (significantly negative, significantly positive or non-significant) of the observed effects is sensitive to how subsidies are modelled: in particular, one-quarter of the models find a significant positive effect of subsidies on technical efficiency, slightly more than half yield a significant negative effect, while the rest have non-significant effects. Some literature shows that results on the impact of subsidy on efficiency and its variability [37] may vary depending on different aspects, such as the time and country reference [38], the specific policy tool and measure [39], the sector of production [40] and other structural and contextual variables, such as the farm size [41].

As far as LFA subsidy is concerned, since its introduction in 1975, the objectives of the measure have evolved, reflecting a changing pattern of social and environmental needs in less-favoured areas and a changing set of priorities [42]. Less-favoured area subsidies are not aimed explicitly at improving productivity, but rather at maintaining production in LFAs, supporting farmers' incomes and contributing to the additional costs of agricultural activities arising from specific handicaps in classified LFAs [43], or favouring the production of specific outputs, such as those in the environmental sphere [36]. However, there is no clear evidence in the literature regarding the effect of LFA and other subsidies on farm incomes, productivity and efficiency, is positive or negative and several studies highlighted that the impact of these payments is limited or controversial, shifting from negative to positive effects depending on countries [44], on economic size [45], and on the temporal perspective, if static or dynamic [46]. The above results put into consideration the effectiveness of area classification and CAP subsidies in stimulating the development of European agriculture.

## 3. Materials and Methods

### 3.1. Data Description and Key Variables

Data from the Italian section of the European Union's Farm Accountancy Data Network (FADN) were used in the study. FADN collects data on income and economic performance of farms in the European Union; among other missions, it is considered a useful tool for European and National policymakers to improve policies affecting territories and farms and, as the rich body of the cited literature demonstrates, it is a good source of information for assessing the efficiency of farms. According to the FADN criteria, the database represents farms that, based on their economic dimension, are considered to be professional and market-oriented and are statistically reliable at the country level in terms of a farm's production system and economic size.

Given the growing importance of the revenues from diversification compared to non-agricultural activities in the formation of farms income, in 2008 a new classification has been introduced in the EU FADN Database, including Other Gainful Activities (OGA) that are directly connected to the agricultural activity and contributes to the formation of farms' income. Among these activities, OGA data includes tourist and recreational activities, the processing of agricultural products and their transformation, the energy production, the aquaculture production, and finally, the contract work performed using farms' assets [47].

In this study, the FADN data have been the basis for a two-stage procedure, comprising: (1) a Data Envelopment Analysis (DEA) at the first stage, followed by (2) a random effects panel Tobit regression (pT) at the second stage.

Table 1 reports variables used in the DEA analysis comprising two output variables, from agricultural and OGA activities, and six input variables, three of which considered structural inputs and the other three being monetary costs.

**Table 1.** Description of variables employed in DEA efficiency analysis.

| Acronyms | Variable Descriptions | Units |
|---|---|---|
| | Output variables | |
| AGROut | Output value from Agricultural Activity | Euro |
| OGAOut | Output value from Other Gainful Activities | Euro |
| | Input variables | |
| UAA | Utilized Agricultural Area | Hectare |
| kW | Machinery Power | kW |
| FWU | Family Work Unit | Units in full time equivalent |
| IntCo | Intermediate Costs | Euro |
| MYCo | Multi-Year Costs | Euro |
| LCo | Labour Costs | Euro |

Table 2 reports indicators employed as independent variables in Tobit analysis. For study purposes, emphasis was given to indicators measuring the farm's performances, as discussed in the literature section. In particular, variables influencing the efficiency of OGA farms have been organised using four sets of variables and indicators: (a) Social variables; (b) Technical-structural variables; (c) Technical-economic variables; (d) and Policy-territorial variables.

**Table 2.** Description of variables and indicators employed in panel Tobit regression.

| Social Variables | |
|---|---|
| **Gender** | **Gender of Farmer** |
| Age | Young (Y/N) farmer according to the EU definition (<40 years) |
| Technical-structural variables | |
| IrrUAA/UAA | Proportion of irrigated UAA |
| kW/UAA | Total power of machinery (kW) per hectare of UAA. It measures the degree of farm mechanization in terms of power available per hectare of surface |
| kW/AWU | Total power of machinery per Annual Work Unit (AWU) in terms of full-time worker; it measures the degree of farm mechanization in terms of available power per work unit |
| UAA/AWU | Utilized Agricultural Area per Annual Work Unit. It measures the intensity of labour use |
| FWU/AWU | Ratio between Family Work Units (FWU) and the total Annual Work Units |
| Technical-economic variables | |
| VA/UAA | Ratio of Value Added (gross output less intermediate inputs) to land used (in Euro). It measures the land productivity |
| VA/AWU | Ratio of Value Added to work units (in Euro). It measures the labour productivity |
| IntCo/TotOut | Total Intermediate Consumption on farm Total Output (in Euro) |
| OGAOut/TotOut | Portion of total output from Other Gainful Activities (OGA) of the farm |
| TEO1 | Farm specialized in crops |
| TEO2 | Farm specialized in livestock |
| TEO3 | Mixed farm |
| Policy-territorial variables | |
| Pill1/TotOut | Subsidies deriving from CAP first Pillar (Direct payments) on total output (in Euro) |
| Pill2/TotOut | Subsidies deriving from CAP second Pillar (Rural development) on total output (in Euro) |
| LFA_1 | Farm located in non-Less Favoured Areas (LFAs) |
| LFA_2 | Farm located in mountain LFAs |
| LFA_3 | Farm located in areas other than mountain LFAs |

Regarding the social variables, this study has taken into consideration "Gender" and "Age"—indeed, if Young or not according to EU definition—attributes of farmers. These characteristics of the manager affect the farm's efficiency and profitability. Gender and age are also considered as some of the factors that significantly affect the decision to diversify activities on the farm, although in both variables, the level of education and the type of diversified activity on farms are considered critical [48,49]. In the panorama of social variables, gender and generation turnover questions are strongly influenced by EU policies, and as such, these variables can be used to indirectly measure the impact of agricultural and rural programmes on the efficiency level of farms.

Among technical-structural variables, the ratio of irrigated land is well considered for use in studying the technical efficiency of farms, as well as their environmental performances [50] in the context of the intrinsic multidimensionality of environmental outcomes [51]. Similarly, many studies highlighted ways in which irrigated farms could have the highest performance [52], although irrigation systems are characterized by higher

inputs cost. Nevertheless, the main irrigated areas are often located in plain and that could represent a limit in terms of chances for farming in marginal lands. One component to measure a farm's efficiency is represented by mechanization; indeed, labour and mechanization are considered two substitutable inputs and their combinations is not just a technical question. Economic efficiency improves by increasing mechanization because it requires less labour in farms, thereby affecting the intensity of labour use; alternately, it could improve labour productivity [53]. For this reason, the variables measuring the total power of agricultural machinery in terms of kW per hectare of UAA and per Work Unit are considered useful to investigate the efficiency of farms. Closely linked to farm mechanization is the intensity of use of labour, above all in specialized agriculture. In farms with diversified activities, this indicator measures the capability to increase the farm efficiency through a better allocation of labour units and time.

As regards Technical-economic variables, the Value Added is often considered in analysis concerning the evaluation of the farm's efficiency, as reported in the literature section. Land (VA/UAA) and labour (VA/AWU) productivities are indicators that provide even more accurate information about the farm's economic performance and are often the basis for the choice between specialized or diversified agriculture.

As reported by FAO [53], intermediate inputs costs (including the cost of purchased feed, breeding, and veterinary services; seeds, fertilizers, and chemicals; repairs, rent, custom hiring, supplies, insurance, gas, oil, and utilities), provide important information on the efficiency of business management, especially in modern agriculture where the weight assumed by intermediate costs is increasingly high.

Concerning the type of farming and productive orientation, we have separately considered two types of specialised farms (crops or livestock) and mixed farms (mixed crop and livestock farms) according to the FADN classification. Mixed farming variable significantly affects the decision to adopt farm income diversification's strategies and leads to a better distribution of family work on the farm [54].

With regards to the political-territorial variables, two types of subsidies have been considered, both CAP direct payments (Pill1/TotOut) and financial supports for rural development (Pill2/TotOut). The value of subsidies received by farms is well considered in the evaluation of their economic performance, as previously discussed. These indicators are also of great political relevance, as subsidies require public justification for spending [55]. Finally, this study has considered the localization of the farms in the territorial systems, i.e., farms located in less-favoured but not in mountain areas (LFA_3), farms in less-favoured mountain areas (LFA_2), and farms not in less-favoured areas (LFA_1), assumed as a benchmark.

*3.2. Methods*

3.2.1. DEA Efficiency Analysis

In the first stage of the study, a DEA analysis was carried out. DEA is a tool for measuring efficiency or performance which has received growing attention in diversified fields of research; within management science, agriculture is among the top latest application fields of DEA [56].

DEA is a nonparametric methodology which produces a single comprehensive measure of performance called efficiency score. Let e_a be the efficiency score of a Decision-Making Unit (DMU)a; it is defined as the ratio between the weighted sums of its outputs and inputs:

$$e_a = \frac{\sum_{r=1}^{s} \mu_{ra} y_{ra}}{\sum_{i=1}^{m} \nu_{ia} x_{ia}} \tag{1}$$

where $x_{ia}$ is the *i*-th input, $y_{ra}$ is the *r*-th output and $\nu_{ia}$, $\mu_{ra}$ are their respective weights. This formulation, together with some compulsory constraints, implies the need to find a solution through a linear fractional problem. A much easier solution turns the problem into a linear programming one, where the objective is $\min(1/e_a)$. If it results equal to 1, then DMU a is efficient, since it is not possible to increase outputs with the same inputs;

otherwise, it is found that $\min(1/e_a) > 1$ and DMU a is inefficient. Output-oriented DEA has been employed, i.e., the linear programme is constructed to determine a firm's potential output given its inputs. In the constant returns to scale model (CRS), inefficient units may become efficient if they proportionally increase all of their outputs, without changing their inputs. Variable returns to scale model (VRS), on the other hand, reflects the fact that production technology may exhibit increasing, constant and decreasing returns to scale. This latter model, compared to the CRS model, shrinks the set of production possibilities, so that a DMU which is efficient under CRS will be efficient as well in the VRS model, but the converse will not necessarily be true. Indeed, the VRS model measures pure technical efficiency free from returns to scale issues, while in the CCR model, the DMU efficiency score might be influenced by scale factors. It is possible to compute the scale efficiency as the ratio between the efficiency scores under the two hypotheses, respectively, $e_{CRS}$ and $e_{VRS}$; for DMU $a$, it is:

$$SE_a = e_{CRS;a}/e_{VRS;a} \text{ , with } SE_a \geq 1.$$

3.2.2. Tobit Regression

In the second stage, VRS DEA efficiency scores obtained in stage 1 have been considered as a dependent variable in the implementation of a random effects panel Tobit regression. This model [57] is employed when there is some form of censoring in the dependent variable, and its general formulation is:

$$y_{it} = \begin{cases} c, & \text{if } y_{it}^* \leq c \\ y_{it}^*, & \text{if } y_{it}^* > c \end{cases} \text{ , with}$$

$$y_{it}^* = x_{it}'\beta + \alpha_i + \nu_{it}$$

(2)

where $x_{it}''$ is the p-dimensional vector of the independent variables observed on unit $i = 1, 2, \ldots, n$ at time t $t = 1, 2, \ldots, T$); $\beta$ is the p-dimensional parametric vector; $\alpha_i$ is the time-independent effect for unit $i$, with $\alpha_i \sim N(0; \sigma_\alpha^2)$ (it is considered as a random variable, from which the name "random effects" is given to the model); and the residual error is $\nu_{it}$, normally distributed with mean 0 and variance $\sigma_\nu^2$ independent of $\alpha_i$. The fit of the model has been assessed through Aldrich-Nelson's pseudo-$R^2$ [58] which is defined as: $R_{AN}^2 = LR/(LR + nT)$, with $LR = 2(l_T - l_0)$, that is, the likelihood ratio statistic, and nT is the total number of observations.

Once the model has been estimated, care must be taken in the interpretation of its parameters; it is not possible to give $\beta$ the same interpretation as in the usual regression model; a correction must be made to the estimated parameters, $\hat{\beta}$, to take into consideration the censoring of the data, the marginal effects on $y_{it}$ is:

$$\frac{\partial E[y_{it}|x_{it}]}{\partial x_{it}} = \hat{\beta} \, \Phi\left(\frac{x_{it}'\hat{\beta}}{\sqrt{\sigma_\alpha^2 + \sigma_\nu^2}}\right)$$

(3)

where $\Phi(.)$ indicates the cdf of the standard normal distribution. In our analysis, we have modelled the log-values of the DEA efficiencies, since it gave us a much better (pseudo-) fit; this means that the constant c in (2) equals zero: $c = 0$.

## 4. Results and Discussion

### 4.1. Descriptive Statistics

In order to appreciate the results of the efficiency and regression analyses, it could be useful to have a look at the main characteristics of farms over all the panel. The panel sample is made up of 305 units with other gainful activities throughout the period 2012–2017, for a total of 1830 observations.

Table 3 reports mean values of the variable used in the DEA analysis (other descriptive statistics are available on request). On average, the output from other activities reaches

about EUR 38,000, and the results are much lower than those derived from the core farming business, but the median amount appears to be quite modest (around EUR 10,000). The high coefficient of variation (CV) for both AGROut and OGAOut confirms a rather heterogeneous distribution of the amount of output values within the sample. The mean UAA of the farms is remarkable (about 55 ha on average) with a CV higher than one.

**Table 3.** Mean values of DEA variables.

| Year | AGROut | OGAOut | UAA | kW | FWU | IntCo | MYCo | LCo |
|------|--------|--------|-----|----|-----|-------|------|-----|
| 2012 | 180,326.3 | 32,263.40 | 53.34 | 283.65 | 1.78 | 62,052.97 | 23,056.65 | 17,274.51 |
| 2013 | 182,616.9 | 37,636.21 | 54.39 | 290.70 | 1.82 | 69,995.03 | 24,239.73 | 19,233.82 |
| 2014 | 175,339.9 | 37,581.81 | 54.02 | 293.63 | 1.81 | 66,326.09 | 24,050.57 | 19,260.84 |
| 2015 | 179,322.6 | 38,363.93 | 54.90 | 295.84 | 1.77 | 62,518.59 | 23,186.14 | 20,082.29 |
| 2016 | 186,828.8 | 41,404.85 | 55.61 | 300.36 | 1.82 | 66,700.42 | 22,634.97 | 21,149.91 |
| 2017 | 195,654.9 | 40,538.23 | 57.17 | 306.14 | 1.80 | 65,997.59 | 21,561.12 | 21,882.70 |
| 2012–2017 | 183,348.20 | 37,964.74 | 54.90 | 295.05 | 1.80 | 65,598.45 | 23,121.53 | 19,814.01 |

Looking at the technical parameter relating to machine power, expressed in kW, it can be noted that sample farms use great power machines with a CV value of one.

As regards labour force, agricultural enterprises in the sample employ less than 2 FWUs on average and even the median value is low and not so much different from the mean, as the CV value (=0.6) shows.

Regarding the cost variables, mean intermediate costs (IntCo) amount to about EUR 66,000, while multi-year costs (MYCo) reach about EUR 23,000 over the period. Labour costs (LCo) amount on average to EUR 20,000, with a low median value (EUR 7600), which probably emphasizes the use of seasonal labour. For all cost variables, median values are far lower than the mean amount, and the coefficients of variation show the heterogeneous distribution of the variables, mainly in the case of intermediate costs borne by farms (with CVs higher than 2).

Regarding the regression analysis, Table 4 reports the descriptive statistics of the dependent variables for 2017; data over the whole period are available in Table S1 (Supplementary Materials), being the DEA efficiency scores used as independent variables. Dependent variables are organized in three main spheres; the technical-structural sphere, to catch the intensity of use of some production factors; the technical-economic sphere, based on some budget indexes; finally, the policy sphere, including the classification of holdings according to the EU less-favoured status as an expression of the territorial context of farms location and two variables measuring the incidence on TotOut of subsidies granted from first and second pillars of the CAP.

**Table 4.** Descriptive statistics of panel Tobit regression variables.

| | Mean | CV | 10% | 25% | 50% | 75% | 90% |
|---|------|----|-----|-----|-----|-----|-----|
| **Continuous Variables** | | | | | | | |
| Technical-Structural Variables | | | | | | | |
| IrrUAA/UAA | 0.28 | 1.42 | 0.00 | 0.00 | 0.00 | 0.60 | 1.00 |
| kW/UAA | 13.80 | 1.33 | 2.10 | 4.10 | 8.79 | 16.06 | 29.53 |
| kW/AWU | 160.36 | 0.95 | 34.49 | 70.00 | 109.17 | 198.46 | 326.80 |
| UAA/AWU | 24.15 | 1.06 | 3.03 | 6.37 | 14.67 | 33.25 | 57.70 |
| FWU/AWU | 0.85 | 0.29 | 0.43 | 0.74 | 1.00 | 1.00 | 1.00 |
| Technical-Economic Variables | | | | | | | |
| VA/UAA | 4811.84 | 1.74 | 548.86 | 895.04 | 2061.23 | 5314.57 | |
| VA/AWU | 41,939.61 | 0.94 | 11,245.80 | 19,745.45 | 33,310.39 | 50,718.37 | 9565.55 |
| OGAOut/TotOut | 0.22 | 1.11 | 0.00 | 0.02 | 0.13 | 0.33 | 78,573.29 |
| IntCo/TotOut | 0.25 | 0.54 | 0.10 | 0.16 | 0.23 | 0.33 | 0.58 |
| Policy Variables | | | | | | | |
| Pill1/TotOut | 0.14 | 2.30 | 0.01 | 0.03 | 0.08 | 0.18 | 0.30 |
| Pill2/TotOut | 0.05 | 1.92 | 0.00 | 0.00 | 0.00 | 0.07 | 0.15 |

**Table 4.** *Cont.*

| | Categorical Variables | |
|---|---|---|
| | | % of Farms |
| Gender | | |
| | Male | 83.48 |
| | Female | 16.52 |
| Age | | |
| | Young | 11.83 |
| | Not Young | 88.17 |
| TEO | | |
| | TEO1 | 50.82 |
| | TEO2 | 31.48 |
| | TEO3 | 17.70 |
| LFA | | |
| | LFA_1 | 42.62 |
| | LFA_2 | 42.30 |
| | LFA_3 | 15.08 |

With reference to the type of farming, expressed according to the EU production orientation categorization (TEO), the distribution of farm units shows a crop specialization (TEO1) in 50.82% of farms, but the weight of output values is lower than that from specialised livestock farms (TEO2 accounting for 31.48% of units in the sample), both in terms of the output value from agricultural activities than from OGA. The location of holdings is considered based on the EU designation of Less Favoured Area (LFA) where the European policy supports farming with difficult contextual and production conditions (Kazimierz et al., 2020). Within the sample, the weight of units located in non-disadvantaged areas is quite similar (LFA_1, 42.62%) compared to that of farms located in mountainous less-favoured areas (LFA_2, 42.30%), the remaining being located in less favoured but not in mountain areas. Regarding gender, 83.48% of holdings in the sample are led by men. As for the age variable, young farmers represent 11.83% of the sample. For the sake of brevity, we refer to Table 4 for the descriptive statistics summarizing the continuous variables used in the panel Tobit regression.

*4.2. DEA Efficiency Results*

Table 5 reports results of farms efficiency based on DEA scores. High margins for efficiency improvement emerged both under the CRS and VRS hypotheses, while the scale efficiency gap is not as relevant; in addition, the variability is quite high for CRS and VRS efficiency scores, and it is different from that of scale efficiency.

**Table 5.** Descriptive statistics of DEA scores, mean values 2012–2017.

| | 10% | 25% | 50% | 75% | 90% | Mean | CV |
|---|---|---|---|---|---|---|---|
| CRS_EFF | 1.019 | 1.250 | 1.742 | 2.351 | 3.140 | 1.936 | 0.449 |
| VRS_EFF | 1.356 | 1.500 | 1.617 | 1.760 | 2.071 | 1.653 | 0.147 |
| Scale_EFF | 1.038 | 1.048 | 1.062 | 1.079 | 1.104 | 1.059 | 0.049 |

Looking at the percentage of efficient units—those reporting a score equal to one—(Figure 1), the weight of efficient units was on average 20.13% under the CRS hypothesis, 25.80% under the VRS hypothesis, and the percentage of farms with scale efficiency was 20.12%. The distribution of units as inefficient or efficient reveals a quite uniform situation by year.

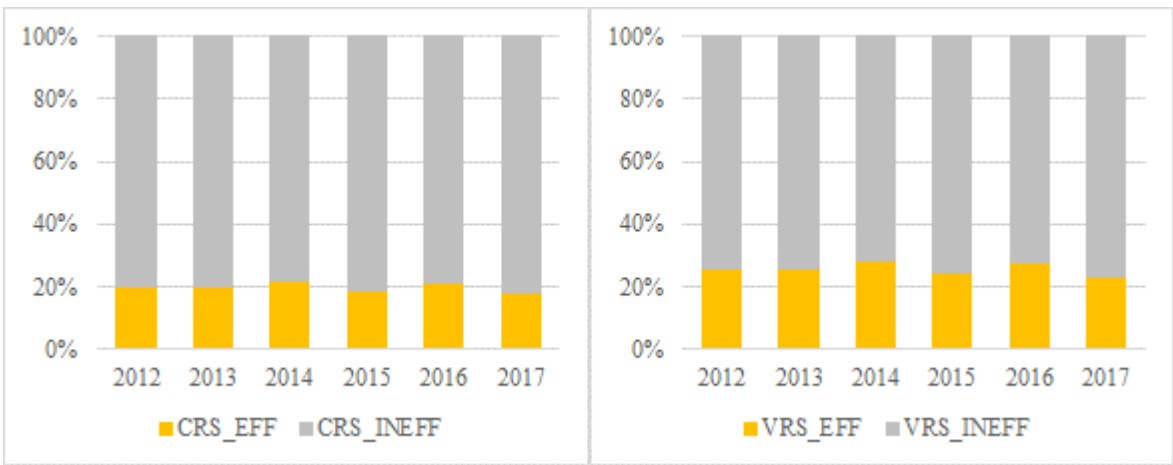

**Figure 1.** Distribution of panel units by year based on DEA (CRS and VRS) scores.

Comparing the mean value of efficiency scores with the median of the same distribution, a positive asymmetry emerges, both under CRS and VRS hypotheses. This implies that the distribution is characterized by the presence of a few units with scores much larger than the others. This is quite a good result considering that the higher the score, the higher is the farm's inefficiency.

Descriptive statistics for the efficient subgroup over the period (Table S2) highlight that the distribution by type of farming does not show any prevalence among the three orientation type classes; in addition, no farms are set in mountain areas; finally, almost all of them are conventional farms, with few organic units. Anyway, it is remarkable that among the efficient units the weight of organic farms increases year by year (from 9.0% in 2012 to 29.6% in 2017). Regarding organic farming, Lakner & Breustedt [59] conclude that organic farms have lower productivity than conventional farms, although, when considering environmental variables, the opposite result could be reported. The output from OGA on average weighs 36% of the total output value; finally, on average over the years, 30.9% of the efficient units were involved in the production of renewable energy almost like other gainful activities traditionally important in Italy, such as agri-tourism [60].

Focusing on the comparison between all farms and inefficient units (Table 6), the inefficiency of farms could be seen by considering changes in the descriptive statistics when compared to those over the whole sample. On average, the inefficiency is measured with a VRS score that changes from 1.88 over the whole sample to 2.18 for the inefficient subsample over years 2012–2017, with a sensible worsening in 2017 (statistics under CRS score are available in Table S3). The coefficient of variation is not very high and just a little bit higher than that overall sample, as expected.

**Table 6.** Descriptive statistics of DEA VRS scores by year—total sample and inefficient units.

| | DEA Scores—All Sample | | | | | |
|---|---|---|---|---|---|---|
| | 2012 | 2013 | 2014 | 2015 | 2016 | 2017 |
| Mean | 1.86 | 1.75 | 1.81 | 1.98 | 1.75 | 2.13 |
| CV | 0.51 | 0.51 | 0.58 | 0.71 | 0.54 | 0.84 |
| 10% | 1.00 | 1.00 | 1.00 | 1.00 | 1.00 | 1.00 |
| 25% | 1.00 | 1.00 | 1.00 | 1.01 | 1.00 | 1.04 |
| 50% | 1.63 | 1.53 | 1.49 | 1.64 | 1.46 | 1.72 |
| 75% | 2.28 | 2.19 | 2.25 | 2.41 | 2.10 | 2.68 |
| 90% | 3.10 | 2.75 | 2.97 | 3.43 | 2.86 | 3.50 |

| | | DEA Scores—Inefficient Units | | | | |
|---|---|---|---|---|---|---|
| | 2012 | 2013 | 2014 | 2015 | 2016 | 2017 |
| Mean | 2.15 | 2.01 | 2.13 | 2.30 | 2.03 | 2.47 |
| CV | 0.44 | 0.45 | 0.51 | 0.65 | 0.47 | 0.77 |
| 10% | 1.18 | 1.16 | 1.12 | 1.19 | 1.14 | 1.20 |
| 25% | 1.50 | 1.41 | 1.38 | 1.50 | 1.39 | 1.45 |
| 50% | 1.91 | 1.80 | 1.89 | 1.91 | 1.85 | 2.05 |
| 75% | 2.58 | 2.41 | 2.60 | 2.62 | 2.38 | 2.99 |
| 90% | 3.44 | 2.98 | 3.14 | 3.59 | 3.12 | 3.76 |

*4.3. Panel Tobit Regression Results*

Table 7 shows the results of the panel Tobit analysis. Between the two demographic variables, only 'Gender (Male)' achieves a significant efficiency (negative value), while the distinction between young and less young farmers is irrelevant. Women may have a higher disposition towards diversification on farms than male farmers, although this is not always evident on the level of performances of the farms themselves [10].

**Table 7.** Second-stage panel Tobit regression results.

| | Estimate | Std. Error | t-Value | Pr (>\|t\|) | Marginal Effects | |
|---|---|---|---|---|---|---|
| (Intercept) | 0.4419 | 0.0933 | 4.736 | 0.0000 | *** | |
| Gender (Male) | 0.1102 | 0.0483 | 2.283 | 0.0224 | * | 0.0822 |
| Young (Yes) | −0.0447 | 0.0354 | −1.263 | 0.2065 | | |
| IrrUAA/UAA | −0.0698 | 0.0401 | −1.739 | 0.0820 | . | −0.0521 |
| kW/UAA | 0.0010 | 0.0013 | 0.709 | 0.4782 | | |
| kW/AWU | 0.0002 | 0.0001 | 1.295 | 0.1955 | | 0.0001 |
| UAA/AWU | −0.0028 | 0.0009 | −3.269 | 0.0011 | ** | −0.0021 |
| FWU/AWU | 0.2665 | 0.0759 | 3.512 | 0.0004 | *** | 0.1989 |
| VA/UAA | 0.0000 | 0.0000 | −8.809 | 0.0000 | *** | 0.0000 |
| VA/AWU | 0.0000 | 0.0000 | −13.300 | 0.0000 | *** | 0.0000 |
| OGAOut/TotOut | −0.2519 | 0.0642 | −3.925 | 0.0001 | *** | −0.1880 |
| IntCo/TotOut | 0.6959 | 0.0815 | 8.538 | 0.0000 | *** | 0.5193 |
| TEO2 | −0.0140 | 0.0416 | −0.337 | 0.7361 | | |
| TEO3 | 0.0385 | 0.0324 | 1.189 | 0.2345 | | |
| LFA_2 | 0.1236 | 0.0511 | 2.420 | 0.0155 | * | 0.0922 |
| LFA_3 | 0.0769 | 0.0689 | 1.117 | 0.2641 | | |
| Pill1/TotOut | 0.0152 | 0.0111 | 1.363 | 0.1727 | | |
| Pill2/TotOut | 0.0472 | 0.0386 | 1.224 | 0.2209 | | |
| *logSigmaMu* | −1.0160 | 0.0510 | −19.940 | 0.0000 | *** | |
| *logSigmaNu* | −1.4050 | 0.0215 | −65.491 | 0.0000 | *** | |

BFGS maximization algorithm. Log-likelihood: −564.7 on 20 df. Signif. codes: 0 '***' 0.001; '**' 0.01; '*' 0.05; '.' 0.1. Aldrich-Nelson pseudo-$R^2$ = 0.336.

Among technical-structural variables, two are highly significant, UAA/AWU and FWU/AWU, but with different signs. UAA/AWU presents a negative sign; this was an expected result since higher labour productivity raises efficiency. The positive sign of FWU/AWU indicates a reduction of the efficiency as family work units increase; this is in line with literature findings [19], which emphasizes how an increase in the incidence of family work is linked to worse performances in terms of efficiency. IrrUAA/UAA is significant, but only at 10% level; the possibility of irrigating increases land profitability [11], and, as a consequence, farm efficiency.

Regarding technical-economic variables, both TEO2 and TEO3 do not have a significant effect from a statistical point of view. The remaining four variables have significant parametric results; however, Value Added/Utilized agricultural areas and Value Added/Work units have negligible impact on the efficiency, in absolute terms. The vari-

able Other gainful activities/Total output has the major positive effect on efficiency, as it reduces both the risks related to price volatility [22], and the agricultural inputs costs [24]. Conversely, the variable Intermediate costs/Total output shows a strong negative influence, since this voice deals with those costs incurred by firms to purchase intermediate inputs, substituting external inputs with internal production factors might promote a corresponding raise in efficiency levels.

The last group—policy and regional variables—includes both CAP 1st and 2nd Pillar subsidies, and the localization of the firms. As observed, both variables referring to CAP are not significant. Indeed, as reported in the literature review, the subsidy-efficiency nexus is ambiguous depending on several aspects and on the applied indicators [61]. In particular, non-significant effects of subsidies can be due to high efficiency scores [62]. De Castris and Di Gennaro [45] demonstrated that public subsidies, while having a positive and significant marginal impact on the added value farms located in Italian lagged regions, switch from negative to positive impacts at higher quantiles, though the intensity of the effects is four times lower than the labour component. Minviel and Sipiläinen [46] show that both in the dynamic and in the static case, public subsidies are negatively associated with farm technical efficiency; nevertheless, these linkages are found to be weak, and they are much weaker when dynamic aspects are taken into consideration; Baležentis et al. [63] showed that production subsidies might be having a negative effect on the efficiency of a family farm in Lithuania. Finally, Biagini et al. [39]) investigated the role of the CAP in enhancing incomes of Italian farms, pointing to differences of effects among policy instruments and across farms; by comparing the different CAP measures affecting farm income, they assessed the very high level of income transfer efficiency for LFAs payments, then of Pillar 2 measures, while the effect of Pillar 1 is not significant.

In this study, the only policy-territorial variable really affecting the efficiency results is the LFA_2 location, which has a significantly lower efficiency than the reference localization in not less-favoured areas. This is most likely because these farms are located in mountainous areas, featured by considerable natural disadvantages, particularly scarce soil productivity or adverse climatic conditions, and in which the preservation of extensive agriculture is important for land management. Indeed, altitude is an aggravating factor for farming, worsening efficiency [64] and profitability [65]. For marginal effects (see Equation (3)), owing to the correction, it was observed that the real impacts of the significant variables are a bit weaker than the corresponding parametric estimates. Anyway, the ranking of the effects is the same, the most negatively impacting variable remains IntCo/TotOut, while OGAOut/TotOut has the highest positive effect on efficiency.

## 5. Conclusions

The paper investigated the efficiency of a panel of Italian farms practising other gainful activities in the period 2012–2017. Compared to previous literature, this study focused on diversified farms and offer insights into their performances and ensuing implications.

Results from the efficiency analysis show that there is much room for improving the performances of diversified farms, even when considering the variability observed among farms in terms of efficiency scores and across the investigated years.

In the second step, panel Tobit findings highlight the positive effects brought by the incidence of output from other gainful activities on the efficiency performance of diversified farms; on the other hand, high intermediate costs have a negative effect on efficiency scores. In addition, regression findings point to the fact that farms located in disadvantaged mountain areas suffer from low efficiency; this result requires further analysis and suggest more consideration to the natural or other area-specific constraints inside the second pillar subsides. In this regard, a particular mention is worth deserving of the effects of CAP subsidies. Although first and second pillar policy effects have proven not to be significant in the regression analysis, it would be nonetheless useful to deepen the investigation on the effects of single policy measures, especially those inside the second pillar, on diversified farms efficiency. Single policy measures could be useful in order to focus on the income

support to disadvantaged areas. Furthermore, the significance of LFA mountain location of farms in terms of their efficiency performance should be more valued in the future framework of the CAP strategic plans at national and regional scales, reducing the gap between farmers' income in non-disadvantaged areas and in LFAs, and also considering their attitude to provide public goods in the socio-environmental sphere. The EU proposal for the next CAP implementation period 2023–2027 allow Member states and regions to consider in their plans income support for disadvantaged areas. Anyway, on one side, it is requested that EU countries ensure that only genuine farmers receive support. On the other hand, the contribution towards climate change objectives of expenditure for natural or other area-specific constraints is only weighted for 40%.

Based on previous comments, study findings suggest future research lines in order to deepen the investigation of factors driving the efficiency of diversified farms for their importance in the agricultural systems. Two main research lines could be of particular interest. First, a research line deepening the analysis by specific types of other gainful activities practised in diversified farms, although, at least for the considered Italian panel sample, the information actually included in the FADN dataset is not so reliable. As previously underlined, a second research line could perform a more detailed analysis of public subsidies by specifically considering single measures, especially those addressed to promote farms diversification phenomena, in order to get new results about the significance of the subsidy-efficiency nexus.

**Supplementary Materials:** The following are available online at https://www.mdpi.com/article/10.3390/su132312949/s1, Table S1: Descriptive statistics of panel Tobit regression variables over the whole period 2012–2017, Table S2: Descriptive statistics of panel Tobit regression variables for the efficient subgroups over the whole period 2012–2017, Table S3: Descriptive statistics of DEA CRS scores by year—Total sample and inefficient units.

**Author Contributions:** Conceptualization, M.B.F., V.G. and L.M.; methodology, L.R. and M.B.F.; formal analysis, L.R.; resources, V.G. and M.B.F.; writing—original draft preparation, M.B.F., V.G., L.M. and L.R.; writing—review and editing, M.B.F., V.G., L.M. and L.R.; visualization, L.R.; supervision, M.B.F. All authors have read and agreed to the published version of the manuscript.

**Funding:** This research received no external funding.

**Institutional Review Board Statement:** Not applicable.

**Informed Consent Statement:** Not applicable.

**Data Availability Statement:** Data was provided by CREA-PB, (Centre for Agricultural Policies and Bioeconomy), Rome, Italy. Crea manages Italian FADN sector.

**Acknowledgments:** We are grateful to Alfonso Scardera of CREA-PB (Rome, Italy) for providing access to the FADN Italian dataset.

**Conflicts of Interest:** The authors declare no conflict of interest.

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
