# Peer review of "Highlighting the Drivers of Italian Diversified Farms Efficiency: A Two-Stage DEA-Panel Tobit Analysis"

_sustainability, doi:10.3390/su132312949_

Round 1

Reviewer 1 Report

Dear authors,

Your paper is good, but I have a few suggestions:

Line 66: close parenthesis.
Lines 78 to 94: review the parenthesis. Perhaps you should perform a thorough review on that.
Line 464: "period, 2012-2017" remove the comma.
Table 6: It is confusing where the "All sample" column stops and "Inefficient units" start. I propose a column for DEA scores on the left

Yours sincerely

Author Response

Line 66: close parenthesis. It has been done.

Lines 78 to 94: review the parenthesis. Perhaps you should perform a thorough review on that. We did it.

Line 464: "period, 2012-2017" remove the comma. It has been done.

Table 6: It is confusing where the "All sample" column stops and "Inefficient units" start. I propose a column for DEA scores on the left. We did it.

Reviewer 2 Report

The topic is interesting and suitable for the scope of the journal. Below are my detailed comments:

Title. Expand DEA.
Line 15. What is FADN data? What is the "two-step" approach?
Line 49. Delete "-" after "payments".
Line 51. Delete "-" after "payments".
Line 66. Add ")".
Line 94. Delete"(".
Line 187. Expand DEA
Table 1. Add column name for the unit.
Table 4. Why do you only provide the descriptive analysis for data of 2017? I suggest you provide the results for all the data used in the regression analysis.
Line 387. I didn't see the results for the subgroups.
Line 400. What's the driver for 2017 to have a high inefficient score.  And you could add CRS results as supplementary material.
Line 415 -422. Have you checked the colinearity between the variables? Because the variables are correlated, it might alter the sign of your coefficient.

Author Response

Title. Expand DEA.

Line 15. What is FADN data? What is the "two-step" approach? We expanded it in Par. 3 “Materials and methods”, the first time we cited them, as we did not consider it appropriate to define it in the abstract.

Line 49. Delete "-" after "payments". It has been done.

Line 51. Delete "-" after "payments". It has been done.

Line 66. Add ")". It has been done.

Line 94. Delete"(". It has been done.

Line 187. Expand DEA. We did it.

Table 1. Add column name for the unit. We did it.

Table 4. Why do you only provide the descriptive analysis for data of 2017? I suggest you provide the results for all the data used in the regression analysis. Many thanks for this suggestion. We have added the results of the descriptive analysis over the whole period in Table S.1 of the Supplementary materials

Line 387. I didn't see the results for the subgroups. We have added Table S.2 in the Supplementary materials

Line 400. What's the driver for 2017 to have a high inefficient score.  And you could add CRS results as supplementary material. We have added the data over the whole period in Table S.3 of the Supplementary materials. Regarding the  reviewer's  comment  about the driver of  the worsening in inefficiency for 2017, the DEA score (for VRS but also for CRS in Table S.3) only tells us about an increase of the inefficient performance of the units, but not about the drivers. In particular, the mean values tell about a general worsening of the performances of farms in the year, and the quantiles show that the problem stays in the farms belonging to the worst two quantiles (70% and 95%), while the median value does not change very much in the period. 

Line 415-422. Have you checked the collinearity between the variables? Because the variables are correlated, it might alter the sign of your coefficient. Yes, of course we checked for multicollinearity issues. None of the VIF indexes exceeds the value of 2; we also calculated the Condition Number, which assumes in our case the very low value of 2.98, clearly indicating no multicollinearity problem.

Round 2

Reviewer 2 Report

I thank the authors' effects in making the edits and adding the supplementary materials. The quality of this manuscript improved.